# Prevalence, pattern and determinants of chronic disease multimorbidity in Nepal: secondary analysis of a national survey

Raja Ram Dhungana ![ORCID],[1] Khem Bahadur Karki,[2] Bihungum Bista,[3] Achyut Raj Pandey,[4] Meghnath Dhimal ![ORCID],[3] Mahesh K Maskey[5]

[1]Nepal Family Development Foundation, Lalitpur, Nepal
[2]Tribhuvan University Institute of Medicine, Maharajgunj, Nepal
[3]Nepal Health Research Council, Kathmandu, Nepal
[4]Abt Associates, Kathmandu, Nepal
[5]Nepal Public Health Foundation, Kathmandu, Nepal

**Correspondence to**
Dr Raja Ram Dhungana;
raja.dhungana@gmail.com

## ABSTRACT

**Objectives** To assess the prevalence, pattern and determinants of non-communicable diseases (NCDs) multimorbidity in Nepal.

**Design** Secondary analysis of the data from the NCD survey 2018, which was conducted between 2016 and 2018.

**Setting** The data belong to the nationally representative survey, that selected the study samples from throughout Nepal using multistage cluster sampling.

**Participants** 8931 participants aged 20 years and older were included in the study.

**Primary outcomes** NCD multimorbidity (occurrence of two or more chronic conditions including hypertension, diabetes, chronic obstructive pulmonary disease, chronic kidney disease, coronary artery disease and cancer). Descriptive statistics, prevalence ratio and odds ratio were computed to assess pattern and determinants of multimorbidity.

**Results** Mean (SD) age was 46.7 years (14.9 years). The majority of the participants were women (57.8%), without formal education (53.4%) and from urban areas (51.5%). Multimorbidity was present in 13.96% (95% CI: 12.9% to 15.1%). Hypertension and diabetes coexisted in 5.7%. Age, alcohol consumption, body mass index, non-high-density lipoprotein (non-HDL) level and rural–urban setting were significantly associated with multimorbidity.

**Conclusion** Multimorbidity was prevalent in particular groups or geographical areas in Nepal suggesting a need for coordinated and integrated NCD care approach for the management of multiplicative co-comorbid conditions.

## BACKGROUND

Globally, non-communicable diseases (NCDs) are giving rise to enormous public health challenges. Recently, the coexistence of two and more NCDs (multimorbidity) has escalated the burden of NCDs globally.[1–6] The prevalence of multimorbidity varies from 12% to 95% across different countries,[7–9] clustering particularly in socioeconomically deprived areas.[10]

Effects of multimorbidity range from an individual to the health system level. In an individual, effects can stretch out from poor physical and mental health outcomes to high treatment burden for the management of multiple chronic diseases.[2 11] Likewise, multiplications of the comorbid conditions could exert enormous pressure on the existing model of healthcare delivery that lacks coordination and integration of NCD services by increasing healthcare costs and workload.[8 11 12]

The major NCDs shared the common behavioural and metabolic risk factors such as physical inactivity, smoking, alcohol consumption and low fruit and vegetables and obesity, which are the leading contributors to burden of multimorbidity.[13 14] Evidence shows that various social circumstances associated with income, education, occupation, gender and ethnicity also determine the distribution of NCDs and multimorbidity.[2 9 15 16] For example, multimorbidity comes about much earlier in life, up to 15 years earlier, among the people belonging to the poor socioeconomic status compared with their wealthier counterparts.[4 17] Additionally, ageing, advancement of healthcare and increasing life expectancy are closely linked with NCDs and thus are also related to multimorbidity.[2 15]

### Strengths and limitations of this study

► This is the first nationally representative study to assess the prevalence, pattern and determinants of non-communicable disease (NCD) multimorbidity in Nepal.
► Data were collected from the national representative sample of 20 years and older adults in Nepal.
► Data relating to cancer are prone to self-reporting bias.
► Estimating the multimorbidity from only six NCDs is likely to underestimate its condition in Nepal.

Nepal, a lower-middle income country in South Asia (landlocked between China and India), hosts a multicultural, multiethnic (over 125 castes/ethnic groups), multilingual (129 first languages) and multireligious (over 10 religions) population of 28 million living mostly in rural areas.[18 19] Recent epidemiological surveys indicated an increasing trend of prevalence of chronic conditions including hypertension (HTN) and diabetes (DM) in Nepal.[20] For example, HTN increased from 21% to 25% between 2007[21] and 2019,[20] and prevalence of DM rose from 3.6% to 5.8% between 2013[22] and 2019.[20] While the prevalence of NCDs is in a rising trend, more individuals are likely to acquire multimorbidity. However, no systematic investigation on NCD multimorbidity is available in Nepal. Detailed epidemiological information is essential for facilitating health system preparedness and response in terms of prevention and management of multimorbidity. Therefore, this study aimed to assess the prevalence, patterns and determinants of multimorbidity in Nepal.

## METHODS
### Data source, study participants and sampling
We used the data from the first nationally representative population-based NCDs survey in Nepal (shortly, NCD survey).[23] The cross-sectional survey was conducted between 2016 and 2018. The purpose of the NCD survey was to estimate the prevalence of selected NCDs including HTN, DM, chronic obstructive pulmonary disease (COPD), chronic kidney disease (CKD), coronary artery disease (CAD) and cancer (CA) in Nepal.

The NCD survey applied a multistage cluster sampling method to select the participants aged 20 years and above from throughout Nepal. Details of the sampling procedure and survey reports are available elsewhere.[23 24] For our study purpose, we included the data from 8931 participants those having information on six NCDs.

### Data collection
The NCD survey collected information on HTN, DM, COPD, CKD, CAD and CA. The survey team recorded three readings of systolic and diastolic blood pressure and used the average of the last two measurements of blood pressure to define HTN as the condition of having systolic blood pressure of ≥140 mm Hg and/or diastolic blood pressure of ≥90 mm Hg and/or the use of antihypertensive medication. The survey considered the participants as diabetics if they had raised fasting glucose (≥126 mg/dL) or raised postprandial blood glucose level (≥200 mg/dL) or the participant was on antidiabetic medication. COPD was diagnosed based on the ratio of $FEV_1$ to FVC and CKD was confirmed using urinary albumin to creatinine ratio and/or measurement of serum creatinine/estimated glomerular filtration rate.[23] For diagnosing CAD, the survey used the (a) information on self-reported admission for myocardial infarction, percutaneous coronary angioplasty or coronary artery bypass surgery; (b) ECG test and (c) responses on the Rose Angina

questionnaire.[25] Details are given in the survey report.[23] CA diagnosis was confirmed as informed by the participants. Details on operational definitions of the study variables are given in the survey report[23] and online supplemental file 1.

The NCD survey also collected information related to sociodemographic characteristics, household income, smoking, alcohol consumption, body mass index (BMI) and lipid. We extracted the data both on the outcome (HTN, DM, COPD, CKD, CAD and CA) and study variables (sociodemographic characteristics, smoking, alcohol consumption, overweight/obesity and high non-high-density lipoprotein (non-HDL)) and defined them as per the survey criteria to assess the burden of NCD multimorbidity in Nepal. Multimorbidity was defined as the presence of two and more NCDs.[10]

### Data analysis
We used the NCD survey information on primary sample units, strata and sampling weight to construct the complex survey weight and performed data analysis using STATA software V.16.1 (Stata Corporation). All estimates are presented with 95% CIs. Geographical distribution and clustering of NCDs are presented graphically. To better understand the independent effects of covariates on multimorbidity, we considered the number of NCDs that occurred in each participant as count data and applied multiple Poisson regression and reported the adjusted relative risk of having a number of NCDs.

For multivariable analysis, we created the hierarchy of the variables as level 1—distant factors (all sociodemographic variables: age, gender, marital status, ethnicity, religion, education, occupation, household income, Province and residence), level 2—intermediate factors (behavioural factors: smoking and alcohol consumption) and level 3—immediate factors (metabolic factors: BMI and non-HDL) and applied hierarchical modelling using a series of stepwise (backward selection, selection criteria: p value <0.2) logistic regression. Hierarchical approach in multivariable data analysis is one of the improved methods for estimating the effect of explanatory variables if there are complex hierarchical inter-relationships between these variables.[26] For sensitivity analysis, we imputed the 568 missing data on non-HDL using the logistic model and replicated the analysis like in the main analysis. A p value <0.05 is considered statistically significant.

### Patient and public involvement
This is the secondary data analysis of the data from the NCD survey. Patients were not involved in the design or conduct of this study, and nor were members of the general public.

## RESULTS
### Sociodemographic characteristics
The mean (SD) age was 46.7 years (14.9 years). The majority of the participants were women (57.8%), married (88.5%), Hindu (88.9%), without formal education (53.4%) and from urban areas (51.5%) (table 1).

**Table 1** Characteristics of the participants (*n*=8931)

| Categories | | Male (%) | Female (%) | Total (%) |
|---|---|---|---|---|
| Age groups | <30 years | 5.83 | 9.51 | 15.34 |
| | 30–44 years | 10.38 | 17.87 | 28.25 |
| | 45–59 years | 15.79 | 16.84 | 32.63 |
| | 60–74 years | 11.48 | 9.12 | 20.60 |
| | ≥75 years | 2.12 | 1.05 | 3.18 |
| Marital status | Never married | 2.75 | 1.85 | 4.60 |
| | Currently married/cohabiting | 41.55 | 46.93 | 88.48 |
| | Separated/divorced | 0.15 | 0.45 | 0.60 |
| | Widowed | 1.15 | 5.17 | 6.32 |
| Ethnicity | Dalit | 3.94 | 5.52 | 9.46 |
| | Disadvantaged Janajatis | 9.45 | 11.29 | 20.74 |
| | Disadvantaged non-Dalit Terai caste | 8.63 | 8.55 | 17.18 |
| | Religious minorities | 1.53 | 1.26 | 2.79 |
| | Relatively advantaged Janajatis | 7.90 | 9.35 | 17.25 |
| | Upper caste groups | 14.16 | 18.43 | 32.59 |
| Religion | Hindu | 40.27 | 48.64 | 88.91 |
| | Buddhist | 2.57 | 2.84 | 5.41 |
| | Muslim | 1.91 | 1.76 | 3.67 |
| | Christian | 0.32 | 0.44 | 0.76 |
| | Kirat and other | 0.54 | 0.71 | 1.25 |
| Education | No education | 17.10 | 36.25 | 53.35 |
| | Primary education | 7.34 | 3.93 | 11.27 |
| | Lower secondary education | 5.22 | 3.42 | 8.64 |
| | Secondary education | 8.31 | 5.67 | 13.98 |
| | Intermediate or plus 2 | 3.59 | 2.95 | 6.54 |
| | Graduate and above | 4.05 | 2.18 | 6.22 |
| Occupation | Government employee | 2.72 | 1.10 | 3.82 |
| | Non-government employee | 2.95 | 1.31 | 4.27 |
| | Self-employed/business | 8.44 | 3.55 | 11.99 |
| | Agriculture (commercial) | 20.99 | 16.83 | 37.82 |
| | Labour | 3.13 | 0.85 | 3.99 |
| | Student | 1.24 | 0.98 | 2.23 |
| | Homemaker | 1.27 | 25.96 | 27.24 |
| | Unemployed | 2.10 | 3.19 | 5.29 |
| | Retired | 2.74 | 0.62 | 3.36 |
| Income quintile | Lowest | 9.78 | 12.47 | 22.25 |
| | Second | 8.59 | 10.38 | 18.96 |
| | Middle | 6.87 | 8.61 | 15.48 |
| | Fourth | 10.99 | 12.29 | 23.29 |
| | Highest | 9.60 | 10.42 | 20.02 |
| Province | Province 1 | 7.49 | 9.30 | 16.78 |
| | Province 2 | 9.37 | 9.03 | 18.40 |
| | Bagmati | 10.60 | 12.98 | 23.58 |
| | Gandaki | 3.92 | 5.51 | 9.43 |
| | Lumbini | 7.91 | 8.97 | 16.88 |
| | Karnali | 2.12 | 3.08 | 5.20 |
| | Sudurpaschim | 4.21 | 5.53 | 9.74 |

Continued

**Table 1** Continued

| Categories | | Male (%) | Female (%) | Total (%) |
|---|---|---|---|---|
| Residence | Rural | 20.97 | 27.55 | 48.52 |
| | Urban | 21.19 | 30.29 | 51.48 |

## Prevalence of NCD multimorbidity

The prevalence of NCD multimorbidity was 13.96% (95% CI: 12.9% to 15.1%). Participants at older ages, men, religious minorities, those without formal education and obese had a higher prevalence of multimorbidity (table 2).

In geographic distribution, Bagmati province had the highest proportion of multimorbidity (figure 1). The prevalence of multimorbidity also varied by urban and rural settings, where more participants from urban than rural had multimorbidity.

## Pattern of multimorbidity

Among the participants having multimorbidity, 80.1% had two NCDs, 17.4% were with three NCDs and 2.5% reported having four and more NCDs. The clustering of NCDs significantly varied with age and gender (figure 2), where NCDs were more likely to occur at old ages and in men (online supplemental file 2).

The most common pair of cooccurrences of NCDs was HTN and DM, followed by HTN and COPD, and HTN and CKD (figure 3). The triad 'HTN, DM and CKD' was present in 1.4%.

## Factors associated with multimorbidity

Age was significantly associated with NCD multimorbidity. The odds of having multimorbidity were 9.4 times for 30–44 years, 27.1 times for 45–59 years and 52.8 for 60–74 years compared with the below 30 years of age group (online supplemental file 3). The probability of occurring multimorbidity was significantly higher in overweight/obesity, alcohol users, urban areas and those who had high non-HDL compared with others (figure 4). The magnitude and direction of the estimates from the imputed dataset were similar to the main analysis.

## DISCUSSION

Our study demonstrated the high burden of NCD multimorbidity in Nepal particularly affecting the elderly, obese, alcohol users, religious minorities, higher-income group and urban residents. Our study reports the very first findings on NCD multimorbidity in Nepal, which may inform the stakeholders on its status, patterns and distribution across Nepal.

The prevalence of NCD multimorbidity in our study was 13.96%. The finding is higher than the rate reported in South Asian countries, India and Pakistan (9.4%),[27] and Bangladesh (8.4%),[28] consistent with the prevalence reported in Canada (12.9%),[13] Switzerland (14.5%)[29] and China (14%)[30] and lower than that of Australia (47.1%),[31] Denmark (39.7%), Brazil (29%),[32] England

**Table 2** Prevalence of multimorbidity by subgroups (n=8931)

| Categories | | Multimorbidity | |
|---|---|---|---|
| | | % | 95% CI |
| Age groups | <30 years | 0.81 | 0.44 to 1.5 |
| | 30–44 years | 6.23 | 5.2 to 7.45 |
| | 45–59 years | 16.75 | 15.1 to 18.54 |
| | 60–74 years | 26.01 | 23.39 to 28.82 |
| | ≥75 years | 39.56 | 32.94 to 46.6 |
| Gender | Female | 12.02 | 10.77 to 13.38 |
| | Male | 16.29 | 14.83 to 17.86 |
| Marital status | Never married | 2.89 | 1.48 to 5.56 |
| | Currently married/cohabiting | 13.9 | 12.79 to 15.09 |
| | Separated/divorced | 14.03 | 6.74 to 26.93 |
| | Widowed | 22.71 | 18.98 to 26.92 |
| Ethnicity | Dalit | 14.26 | 11.77 to 17.17 |
| | Disadvantaged Janajatis | 11.4 | 9.79 to 13.24 |
| | Disadvantaged non-Dalit Terai caste | 14.69 | 12.36 to 17.38 |
| | Religious minorities | 21.14 | 15.98 to 27.41 |
| | Relatively advantaged Janajatis | 18.45 | 15.52 to 21.79 |
| | Upper caste groups | 12.14 | 10.64 to 13.83 |
| Religion | Hindu | 13.66 | 12.54 to 14.87 |
| | Buddhist | 15.29 | 11.53 to 20 |
| | Muslim | 20.06 | 15.49 to 25.58 |
| | Christian | 9.34 | 4.17 to 19.63 |
| | Kirat and others | 14.49 | 8.25 to 24.21 |
| Education | No education | 15.85 | 14.45 to 17.36 |
| | Primary education | 14.32 | 11.9 to 17.13 |
| | Lower secondary education | 10.8 | 8.38 to 13.81 |
| | Secondary education | 11.91 | 9.72 to 14.52 |
| | Intermediate or plus 2 | 10.99 | 8.15 to 14.66 |
| | Graduate and above | 9.28 | 6.87 to 12.42 |
| Occupation | Government employee | 16.69 | 12.29 to 22.26 |
| | Non-government employee | 12.79 | 9.07 to 17.75 |
| | Self-employed/business | 15.98 | 12.95 to 19.56 |
| | Agriculture (commercial) | 11.78 | 10.37 to 13.34 |
| | Labour | 10.23 | 7.36 to 14.03 |
| | Student | 0.37 | 0.09 to 1.47 |
| | Homemaker | 14.81 | 13.01 to 16.83 |
| | Unemployed | 18.27 | 14.35 to 22.98 |
| | Retired | 29.71 | 24.55 to 35.45 |
| Income quintile | Lowest | 11.5 | 9.91 to 13.32 |
| | Second | 11.73 | 9.97 to 13.75 |
| | Middle | 13.2 | 11.16 to 15.55 |
| | Fourth | 16.96 | 14.83 to 19.32 |
| | Highest | 15.87 | 13.75 to 18.24 |
| Residence | Rural | 10.89 | 9.69 to 12.23 |
| | Urban | 17.03 | 15.32 to 18.89 |
| Smoking (current) | No | 13.9 | 12.71 to 15.19 |
| | Yes | 14.17 | 12.37 to 16.19 |

Continued

**Table 2** Continued

| Categories | | Multimorbidity | |
|---|---|---|---|
| | | % | 95% CI |
| Alcohol consumption (current) | No | 13.28 | 12.11 to 14.56 |
| | Yes | 15.91 | 14.08 to 17.92 |
| Weight | Underweight (<18.5 kg/m$^2$) | 12.26 | 9.82 to 15.21 |
| | Normal (18.5–24.9 kg/m$^2$) | 10.91 | 9.87 to 12.05 |
| | Overweight (25–29.9 kg/m$^2$) | 20.74 | 18.32 to 23.38 |
| | Obese (≥30 kg/m$^2$) | 30.28 | 25.48 to 35.55 |
| High non-HDL | No (<130 mg/dL) | 11.5 | 10.4 to 12.7 |
| | Yes (≥130 mg/dL) | 18.73 | 16.87 to 20.75 |

Non-HDL, Non-high-density lipoprotein.

(27.2%)[3] and Vietnam (16.4%).[33] A systematic review that computed a pooled estimate of multimorbidity from 18 high-income countries and 31 low/middle-income countries had found the prevalence of multimorbidity to be 33.1%.[34] As the prevalence of chronic conditions tends to differ across different countries owing to differences in dietary practices, lifestyle, alcohol consumption, tobacco use and level of physical activity, among others, it is not uncommon for multimorbidity to vary across countries. Some of these disparities could be because of the differences in the methodology used in individual study. For example, prevalence of multimorbidity also depends on the number of diseases considered in the study, population under study and selection of participants.

The most common pair of cooccurrences of NCDs was HTN and DM (5.7%). The prevalence of comorbid DM and HTN was two times as high as that we reported (2%) in 2013 in Nepal, suggesting an increasing burden of DM and HTN comorbidity over the years.[35] The prevalence of comorbid DM and HTN was found to be 4.5% in India[36] and 3% in Bangladesh.[37] Similarly, a study by Harrison *et al* in Australia had found that the cooccurrence of HTN and DM was 4.1%.[31] The higher cooccurrence of DM and HTN or the comorbid condition could be because both conditions are more prevalent in society and also share common risk factors.

The prevalence of multimorbidity was found to be slightly higher among men in our study. The prevalence of multimorbidity was 16% in men and 12% in women.

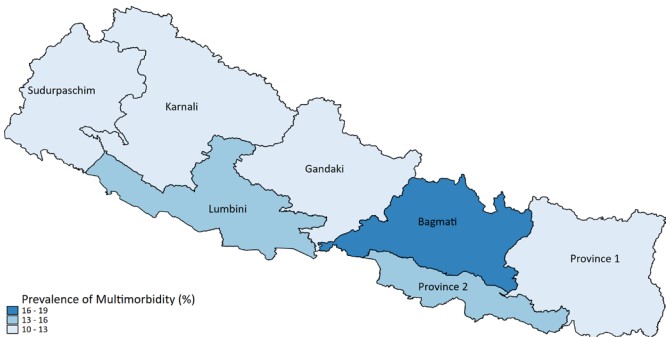

**Figure 1** Distribution of multimorbidity by Province.

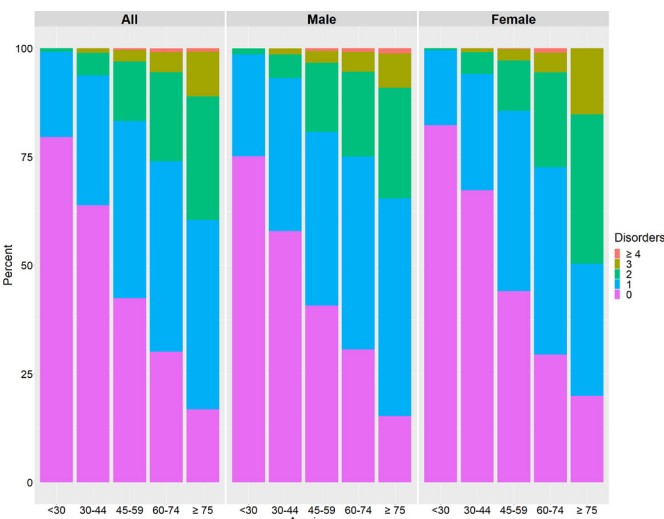

**Figure 2** Clustering of NCDs by gender. Note: the prevalence of NCDs was likely to increase with ageing and was comparatively higher in men than women. Interestingly, compared with the men of the same age group, women of age 45 years and younger were less likely to have NCDs, whereas women of age ≥75 years had higher prevalence of living with at least two NCDs (multimorbidity). NCDs, non-communicable diseases.

After adjusting age, the difference in prevalence between men and women was not significant. Like our finding, one of the systematic reviews showed no statistically significant difference in the pooled prevalence of multimorbidity by gender though the nine of the studies that were reviewed reported a higher rate of multimorbidity in women than in men.[9] Likewise, the prevalence of multimorbidity was 14.8% in men and 14.3% in women in Switzerland,[29] and 17.2% in men and 15.5% in women in Vietnam, indicating no disparity in multimorbidity in-between.[33]

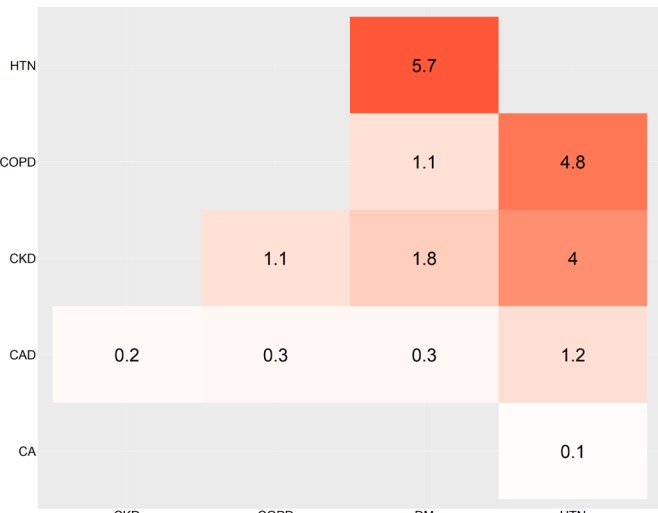

**Figure 3** Cooccurrence of NCDs. Cooccurrence of six major NCDs given in per cent. CA, cancer; CAD, coronary artery disease; CKD, chronic kidney disease; COPD, chronic obstructive pulmonary disease; DM, diabetes; HTN, hypertension; NCDs, non-communicable diseases.

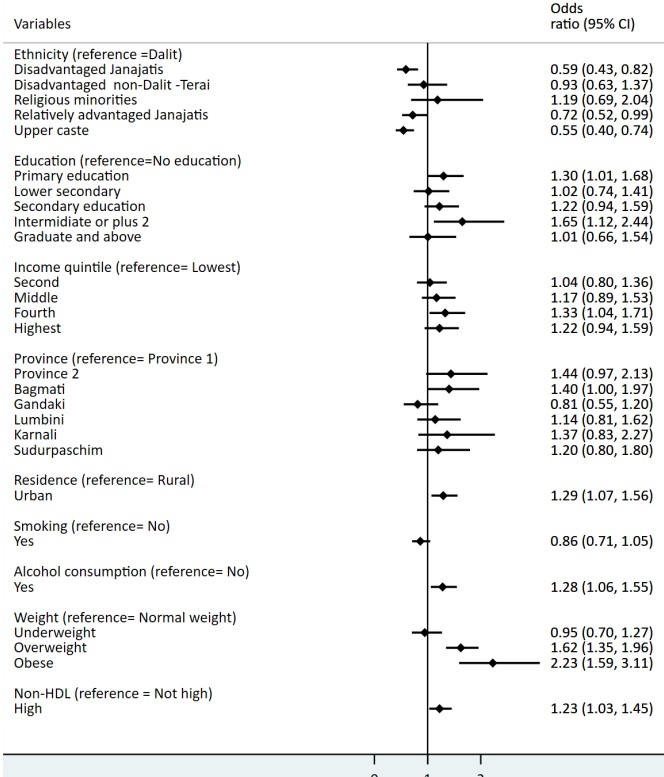

**Figure 4** Factor associated with multimorbidity. OR was estimated from the hierarchical logistic regression models. Details of output are given in online supplemental file 2. Non-HDL, non-high-density lipoprotein.

Multiple studies conducted in different setting have revealed age as a determinant of multimorbidity with the condition being more common with increasing age group.[5 29 38] The prevalence of multimorbidity was 0.81% among participants of age below 30 years and 39.5% among participants of age 75 years or above. In a similar study in Australia, the prevalence of multimorbidity was found to be 1.7% in participants of age 20–29 years, 33.6% among participants of age 70–79 years and 37.7% in participants of age 80 years which also indicates that the multimorbidity increases with increasing age.[29] In another study in the USA, the prevalence of multimorbidity was found to be 67% among those above 65 years and 71% among those above 75 years of age.[39] As most of the NCDs are more common in the older age group, there could have been a higher probability of cooccurrence and thus leading to multimorbidity.

We found that urban area and Bagmati province (capital) had a higher prevalence of multimorbidity. This could be linked to the high burden of NCD risk factors in those areas. Recent surveys suggested that smoking, alcohol consumption, physical inactivity, high salt intake and obesity were more prevalent in Bagmati province and urban areas than others.[20 40] Likewise, alcohol consumption, one of the well-established NCD risk factors was also significantly associated with multimorbidity.

In our study, the odds of having multimorbidity was approximately threefolds higher in obese compared with

normal-weight participants. Similar findings were also reported in studies by Agborsangaya et al[14] and Khan et al,[28] where the odds of having multimorbidity was twofolds higher in obese. A higher risk of multimorbidity among the obese population could be because it is one of the most common risk factors shared among NCDs. This study also found that high non-HDL cholesterol, a strong predictor of cardiovascular diseases,[41] was also significantly associated with multimorbidity.

This is the first study to assess the burden of multimorbidity in Nepal. Findings from the nationally representative study could have a high policy implication in terms of tackling the growing burden of multimorbidity in Nepal. Our study found that NCD multimorbidity is prevalent in Nepal. Higher prevalence of multimorbidity means more frequent hospital consultations and more hospitalisations.[3 5 6] For example, Schneider et al[6] reported that the patients with three or more chronic conditions use 25 times more hospital beds than peers without any chronic conditions. As higher proportion of healthcare encounters is likely to occur because of multimorbidity, appropriate training of the health facilities at peripheral level and development of an effective referral system could be useful in resource constraint setting like Nepal.

Likewise, with the growing burden of multimorbidity, health system also needs to prepare for dealing with an additional number of health facility visits in Nepal. The current health system organisation largely focuses on preventing and managing maternal and child health conditions. It requires a degree of reorganisation to deal with the increasing burden of NCDs and multimorbidity. There are several evidence-based models which are in practice for the management of multimorbidity globally. Sustainable integrated chronic care models for multimorbidity: delivery, financing and performance is an example of an innovative framework that suggests the integration of care at micro-level, meso-level and macro-level according to six pillars of health system: service delivery, leadership and governance, workforce, financing, technologies and medical products and information and research.[42] The strategies, however, should be specific to the Nepalese context and backed by evidence on health benefit and cost-effectiveness as the most common comorbid conditions could vary from one country to another.

This study has some limitations. The use of self-reported data for CA diagnosis might have underestimated the prevalence of CA in the original study, affecting the current study findings. Likewise, our study could not include other several comorbid conditions such as mental disorders and degenerative diseases and possibly underestimated the burden of multimorbidity in Nepal. For that, a national representative survey encompassing the majority of the common NCDs is recommended.

## CONCLUSIONS

This study demonstrated that NCD multimorbidity is prevalent and disproportionately distributed across different socioeconomic and demographic strata in Nepal. Growing prevalence of multimorbidity indicates an urgency for the health system to prepare and respond to the inevitable health threats and economic burden in Nepal.

**Acknowledgements** We would like to thank Nepal Health Research Council for sharing the data. We appreciate all the participants for their great contribution to the study by sharing the information.

**Contributors** RRD and KBK conceptualised the study. RRD analysed the data. RRD, BB and ARP interpreted the findings and prepared the first draft of the manuscript. KBK, MD and MKM interpreted the findings and revised the first draft of the manuscript. All authors read and approved the final manuscript. KBK is the co-first author.

**Funding** The authors have not declared a specific grant for this research from any funding agency in the public, commercial or not-for-profit sectors.

**Map disclaimer** The inclusion of any map (including the depiction of any boundaries therein), or of any geographic or locational reference, does not imply the expression of any opinion whatsoever on the part of BMJ concerning the legal status of any country, territory, jurisdiction or area or of its authorities. Any such expression remains solely that of the relevant source and is not endorsed by BMJ. Maps are provided without any warranty of any kind, either express or implied.

**Competing interests** None declared.

**Patient consent for publication** Not required.

**Ethics approval** The NCD survey obtained ethical approval from the Ethical Review Board of Nepal Health Research Council (ethical approval ID: nr. 110/2016). Informed consent was obtained from the study participants ensuring voluntary participation, privacy and confidentiality.

**Provenance and peer review** Not commissioned; externally peer reviewed.

**Data availability statement** Data used in the study are reposited in Nepal Health Research Council and are available from the council upon a reasonable request to the Council.

**ORCID iDs**
Raja Ram Dhungana http://orcid.org/0000-0002-9610-6306
Meghnath Dhimal http://orcid.org/0000-0001-7176-7821

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
