## [Reviewer comments · BMJ Open]

ARTICLE DETAILS

TITLE (PROVISIONAL)	Prevalence, pattern, and determinants of chronic disease multimorbidity in Nepal: secondary analysis of a national survey
AUTHORS	Dhungana, Raja Ram; Karki, Khem; Bista, Bihungum; Pandey, Achyut; Dhimal, Meghnath; Maskey, Mahesh

VERSION 1 – REVIEW

REVIEWER	Birtwhistle, Richard Queens University, Family Medicine
REVIEW RETURNED	28-Jan-2021

GENERAL COMMENTS	This paper reports a secondary analysis of a national survey of noncommunicable disease in Nepal conducted between 2016-2018. The authors describe this as a nationally representative survey with multistage cluster sampling. The survey included 8931 participants aged 20 years and older who were interviewed, had blood pressure measurements done as well as some laboratory testing for diabetes, chronic kidney disease, COPD and an ECG and Rose questionnaire for coronary artery disease. The interviewers also depended on participant self-reporting for CAD and cancer. Other chronic diseases were not determined. Information was collected related to socio-demographic characteristics, household income, smoking, alcohol consumption, body mass index (BMI) and lipid. Data analysis included descriptive statistics, concentration indices for health and economic inequality and multivariable analysis for Sociodemographic and risk factor variables. All participants gave informed consent and the study had ethics approval. Mean age was 46.7 years. 57.8% of the participants were female, without formal education and from urban areas. Multimorbidity was present in 13.96%. Hypertension and diabetes coexisted in 5.7%. Prevalence of multimorbidity, hypertension and diabetes was higher in higher income groups. Age, alcohol consumption, body mass index, non-HDL level and rural-urban setting were significantly associated with multimorbidity. I found this an interesting description of chronic disease in Nepal and the authors suggest this is the first such description. The analysis seems appropriate and the discussion of the results is good. As the authors suggest the number of chronic diseases was limited and did not include diseases like osteo arthritis or any mental health disorders. To improve this paper, it will need editing to correct grammatical errors and typos (eg. Diastolic pressure > 190 mmHg). Also it would be helpful for the reader to know more about the context of Nepal in relation to geography and social demography (ethnicity, religion, education, employment and income). I think that
--

	specific information about religious and ethnic groups should be removed or the authors should explain why it is important for this paper and put it in some context. This paper reports a secondary analysis of a national survey of noncommunicable disease in Nepal conducted between 2016-2018. The authors describe this as a nationally representative survey with multistage cluster sampling. The survey included 8931 participants aged 20 years and older who were interviewed, had blood pressure measurements done as well as some laboratory testing for diabetes, chronic kidney disease, COPD and an ECG and Rose questionnaire for coronary artery disease. The interviewers also depended on participant self-reporting for CAD and cancer. Other chronic diseases were not determined. Information was collected related to socio-demographic characteristics, household income, smoking, alcohol consumption, body mass index (BMI) and lipid. Data analysis included descriptive statistics, concentration indices for health and economic inequality and multivariable analysis for Sociodemographic and risk factor variables. All participants gave informed consent and the study had ethics approval. Mean age was 46.7 years. 57.8% of the participants were female, without formal education and from urban areas. Multimorbidity was present in 13.96%. Hypertension and diabetes coexisted in 5.7%. Prevalence of multimorbidity, hypertension and diabetes was higher in higher income groups. Age, alcohol consumption, body mass index, non-HDL level and rural-urban setting were significantly associated with multimorbidity. I found this an interesting description of chronic disease in Nepal and the authors suggest this is the first such description. The analysis seems appropriate and the discussion of the results is good. As the authors suggest the number of chronic diseases was limited and did not include diseases like osteo arthritis or any mental health disorders. To improve this paper, it will need editing to correct grammatical errors and typos (eg. Diastolic pressure > 190 mmHg). Also it would be helpful for the reader to know more about the context of Nepal in relation to geography and social demography (ethnicity, religion, education, employment and income). I think that specific information about religious and ethnic groups should be removed or the authors should explain why it is important for this paper and put it in some context.
--	---

REVIEWER	Khan, Nusrat BRAC, Research and Evaluation Division, BRAC
REVIEW RETURNED	24-Feb-2021

GENERAL COMMENTS	The study is very timely given the rising prevalence of multimorbidity globally and lack in South Asian country data. As from the national survey design, the strength of the study is its representativeness and major limitation is number of diseases; as there is strong evidence of linearity with number of diseases and prevalence of multimorbidity. Below are the comments to be addressed:  -Pattern and distribution: are you referring to clustering of diseases? If so, pattern is more appropriate as various determinants are later stratified. -What is the explanation for using High non-HDL as an exposure variable? -The paper needs to keep a primary focus of reporting; e.g. Income
---

	quintile/lifestyle variables. For income quintile rationale behind using concentration index needs to be further explained.  -Statistical analysis: why hierarchal model was developed? This is not essential for multivariate regression modelling. For the components of the 3 hierarchy, are those self identified or any standard guideline is used? -Figure 2 needs to be explained further. -Effect size and goodness of fit should be checked before advanced analysis - In discussion section, it needs to be more coherent in terms of outcomes; e.g. socioeconomic factors, lifestyle factors and metabolic factors along with references from South Asian population. - Disease and disorder have been used interchangeably as terminology, which needs to be consistent. Overall, a good study which will shed light on the status of multimorbidity on this under studied population.
--	---

VERSION 1 – AUTHOR RESPONSE

Reviewer: 1

Dr. Richard Birtwhistle, Queens University

Comments to the Author:

This paper reports a secondary analysis of a national survey of noncommunicable disease in Nepal conducted between 2016-2018. The authors describe this as a nationally representative survey with multistage cluster sampling. The survey included 8931 participants aged 20 years and older who were interviewed, had blood pressure measurements done as well as some laboratory testing for diabetes, chronic kidney disease, COPD and an ECG and Rose questionnaire for coronary artery disease. The interviewers also depended on participant self-reporting for CAD and cancer. Other chronic diseases were not determined. Information was collected related to socio-demographic characteristics, household income, smoking, alcohol consumption, body mass index (BMI) and lipid. Data analysis included descriptive statistics, concentration indices for health and economic inequality and multivariable analysis for Sociodemographic and risk factor variables.

All participants gave informed consent and the study had ethics approval.

Mean age was 46.7 years. 57.8% of the participants were female, without formal education and from urban areas. Multimorbidity was present in 13.96%. Hypertension and diabetes coexisted in 5.7%. Prevalence of multimorbidity, hypertension and diabetes was higher in higher income groups. Age, alcohol consumption, body mass index, non-HDL level and rural-urban setting were significantly associated with multimorbidity.

I found this an interesting description of chronic disease in Nepal and the authors suggest this is the first such description. The analysis seems appropriate and the discussion of the results is good.

As the authors suggest the number of chronic diseases was limited and did not include diseases like osteo arthritis or any mental health disorders. To improve this paper, it will need editing to correct grammatical errors and typos (eg. Diastolic pressure > 190 mmHg). Also it would be helpful for the reader to know more about the context of Nepal in relation to geography and social demography (ethnicity, religion, education, employment and income).

Response: Thank you for your comment. We have now added some background information about Nepal in the introduction - as follows

“Nepal, a lower-middle-income country in South Asia (landlocked between China and India), hosts a multi-cultural, multi-ethnic (over 125 caste/ethnic groups), multi-lingual (129 first languages) and multi-

religious (over 10 religions) population of 28 million living mostly in rural areas [18 19].”

Regarding grammatical errors, we have reviewed the manuscript and corrected as required. We will also let you know that the manuscript will be sent for language editing in the final round revision.

Comment: I think that specific information about religious and ethnic groups should be removed or the authors should explain why it is important for this paper and put it in some context.

Response: We discussed your concern about omitting religion and ethnicity from the analysis. All authors (including three of those from the Ministry of Health), however, emphasized the importance of religion and ethnicity-specific estimates in Nepal public health program planning and implementation as the distribution and determinants of diseases significantly vary by ethnic and religious cluster in Nepal. Therefore, we decided to report the estimates based on ethnicity and religion, which is in line with the original report (NCD surveys), national census and other Nepalese national surveys. To give more information on the ethnic and religious background of Nepal, we have now added the following information in the introduction –

“.....hosts a multi-cultural, multi-ethnic (over 125 caste/ethnic groups), multi-lingual (129 first languages) and multi-religious (over 10 religions) population of 28 million living mostly in rural areas [18 19]...”

Reviewer: 2

Dr. Nusrat Khan, BRAC

Comments to the Author:

The study is very timely given the rising prevalence of multimorbidity globally and lack in South Asian country data. As from the national survey design, the strength of the study is its representativeness and major limitation is number of diseases; as there is strong evidence of linearity with number of diseases and prevalence of multimorbidity.

Comments: Below are the comments to be addressed:

-Pattern and distribution: are you referring to clustering of diseases? If so, pattern is more appropriate as various determinants are later stratified.

Response: Thank you for your suggestion. We have now removed the word “distribution” from the title.

Comment: What is the explanation for using High non-HDL as an exposure variable?

Response: As the recent evidence (some of the links are given below) suggested that non-HDL is a better risk predictor of Small-Dense LDL Cholesterol (bad cholesterol) and cardiovascular diseases than other cholesterol, we used it in our model.

<https://www.ncbi.nlm.nih.gov/pmc/articles/PMC5090818/>

<https://www.sciencedirect.com/science/article/pii/S014067361932519X>

<https://academic.oup.com/clinchem/article/57/3/490/5621018?login=true>

Comment: The paper needs to keep a primary focus of reporting; e.g. Income quintile/lifestyle variables. For income quintile rationale behind using concentration index needs to be further explained.

Response: Thank you for your suggestion. We have now removed the information related to concentration index from the manuscript to focus the findings on prevalence, pattern and determinants of multimorbidity.

Comment: Statistical analysis: why hierarchal model was developed? This is not essential for

multivariate regression modelling. For the components of the 3 hierarchy, are those self identified or any standard guideline is used?

Response: We understand your concern about hierarchical logistic regression. Hierarchical logistic regression is commonly used to adjust/ partition out the effect of background variables in the model. We could have included all variable in the model in our study rather than stepwise hierarchical logistic regression. However, using hierarchical logistic regression would generate more reproducible (due to better variable selection method when we have different layers of several variables) and precise results (controlling multicollinearity) than only stepwise regression/ all variable in the model. Detailed discussion is available here <https://files.eric.ed.gov/fulltext/ED534385.pdf>, <https://www3.nd.edu/~rwilliam/stats1/x95.pdf> , <http://web.mnstate.edu/malonech/Psy633/Notes/Hierarchical%20vs%20Step-wise.doc> .

Regarding three components/hierarchy of the variables, socio-demographic, behavioural and metabolic risk factors were grouped within the cluster of “distant”, “intermediate” and “immediate” risk factors and entered in the model in respective order (starting with distant). The reason for doing so is - despite their independent association with NCDs, they might also follow the pathway as shown in the figure attached separately. Entering the cluster of variables separately (in hierarchical order) in the model could help us select the variables (as each model contains a small number of variable compared to all variable at once/stepwise model), detect collinearity or interaction easily (by assessing the change in beta coefficient in each step) and check the model fit (comparing AIC, BIC in each step).

-Figure 2 needs to be explained further.

Response: Following information is now added in the note of figure two:

“Note: The prevalence of NCDs was likely to increase with aging and was comparatively higher in males than females. Interestingly, compared to the males of the same age group, females of age 45 years and younger were less likely to have NCDs, whereas females of age ≥ 75 years had a higher prevalence of living with at least two NCDs (multimorbidity).

Comment: Effect size and goodness of fit should be checked before advanced analysis

Response: Cross tabulation and univariate odds ratio were calculated before constructing the final model. The information/table was not included in the manuscript as they were redundant to table 2 (where we have given 95% CI), and supplementary file 3 (where we have given an adjusted odds ratio). However, they are now uploaded separately for your review.

Comment: In discussion section, it needs to be more coherent in terms of outcomes; e.g. socioeconomic factors, lifestyle factors and metabolic factors along with references from South Asian population.

Response: Thank you for your suggestion. We have now completely revised the discussion in the order of socio-demographic, behavioural and metabolic factors and compared with the South Asian population where available. Examples of revision are given below

“.....The prevalence of NCD multimorbidity in our study was 13.96%. The finding is higher than the rate reported in South Asian counterparts, India and Pakistan (9.4%) [27], and Bangladesh (8.4%) [28], consistent with the prevalence reported in Canada (12.9%) [13].....”

“.....We found that urban area and Bagmati province (capital) had a higher prevalence of multimorbidity. This could be linked to the high burden of NCD risk factors in those areas. Recent surveys suggested that smoking, alcohol consumption, physical inactivity, high salt intake, and obesity were more prevalent in Bagmati province and urban areas than others [20 40]. Likewise,

alcohol consumption, one of the well-established NCD risk factors was also significantly associated with multimorbidity.

“...In our study, the odds of having multimorbidity was approximately 3 folds higher in obese compared to normal-weight participants. Similar findings were also reported in studies by Agborsangaya et al. [14] and Khan et al. [28] where odds of having multimorbidity was 2 folds higher in obese. Higher risk of multimorbidity among the obese population could be because it is one of the most common risk factors shared among the NCDs. This study also found that the high non-HDL cholesterol, a strong predictor of cardiovascular diseases [41], was also significantly associated with multimorbidity. ...”

Comment: Disease and disorder have been used interchangeably as terminology, which needs to be consistent.

Response: Revised

Comment: Overall, a good study which will shed light on the status of multimorbidity on this under studied population.

Response: Thank you for your positive remarks.

VERSION 2 – REVIEW

REVIEWER	Birtwhistle, Richard Queens University, Family Medicine
REVIEW RETURNED	02-Jun-2021

GENERAL COMMENTS	The revised version of the paper is well done and reviewer suggestions thoughtfully considered. I have no further suggestions for revision.
---

REVIEWER	Khan, Nusrat BRAC, Research and Evaluation Division, BRAC
REVIEW RETURNED	14-Jun-2021

GENERAL COMMENTS	After the revision, the manuscript meets the standards of publication o BMJ open. Need some explanation on why hierarchical logistic regression model was used instead of regular/ stepwise model. Few spelling corrections are needed. Figure 2 and 3 in the supplementary section can be improved for better visualization and needs better resolution quality.
---

VERSION 2 – AUTHOR RESPONSE

Comment:

After the revision, the manuscript meets the standards of publication o BMJ open. Need some explanation on why hierarchical logistic regression model was used instead of regular/ stepwise model. Few spelling corrections are needed. Figure 2 and 3 in the supplementary section can be improved for better visualization and needs better resolution quality.

Response: Thank you for your comments. We have now added an explanation for using a hierarchical logistic regression model with a relevant citation in the manuscript.

“For multivariable analysis, we created the hierarchy of the variables as level 1 - distant factors (all

socio-demographic variables: age, gender, marital status, ethnicity, religion, education, occupation, household income, Province, and residence), level 2 -intermediate factors (behavioural factors: smoking and alcohol consumption) and level 3- immediate factors (metabolic factors: body mass index and non-HDL) and applied hierarchical modelling using a series of stepwise (backward selection, selection criteria: p-value <0.2) logistic regression. Hierarchical approach in multivariable data analysis is one of the improved methods for estimating the effect of explanatory variables if there are complex hierarchical inter-relationships between these variables [26].”

Figure 2 and 3 are now revised. The pdf versions of supplementary files have now been replaced by an excel sheet for better quality.